# Effects of Exogenous Organic Acids on the Soil Metabolites and Microbial Communities of *Panax notoginseng* from the Forest Understory

**Jingying Hei** [1,2,†], **Yue Li** [1,2,†], **Qiong Wang** [2,†], **Shu Wang** [1,*] and **Xiahong He** [2]

[1] Key Laboratory of In-Forest Resource Protection and Utilization in Yunnan Province, Southwest Forestry University, Kunming 650224, China; h2940794807@163.com (J.H.); liyue_421@163.com (Y.L.)

[2] Key Laboratory for Forest Resources Conservation and Utilization in the Southwest Mountains of China, Southwest Forestry University, Ministry of Education, Kunming 650224, China; millmeng@swfu.edu.cn (Q.W.); hxh@swfu.edu.cn (X.H.)

[*] Correspondence: wangshu@swfu.edu.cn; Tel.: +86-871-6386-3632

[†] These authors contributed equally to this work.

**Abstract:** *Panax notoginseng* (Sanqi) is a precious traditional Chinese medicine that is commonly cultivated using conventional management methods in agricultural systems in Yunnan, China, where it faces the challenge of continuous cropping obstacles (CCOs). However, the existence of Sanqi CCOs in Sanqi–pine agroforestry systems remains unclear. Here, we applied three types of exogenous organic acids (phthalic acid, palmitic acid, and phthalic + palmitic mixed organic acids) mainly derived from the root exudates of Sanqi to simulate the CCOs; then, we compared the effects on plant growth, soil physicochemical properties, soil microbes, and soil metabolites. We found that organic acid concentrations of >250 mg/kg reduced the fresh weight of Sanqi and the levels of total nitrogen, ammonium nitrogen, soil water content, total phosphorus, and pH value; these concentrations also increased the soil bacterial and fungal α-diversity. The type of organic acid, as opposed to the concentration and duration of treatment, had a vital impact on the structure of the bacterial and fungal community in Sanqi soils. Moreover, the organic acid concentrations of >250 mg/kg also decreased the complexity and stability of the bacterial and fungal network. In addition, the metabolic pathways in the soils under different organic acids included 17 differential metabolites (DMs), which were involved in steroid hormone biosynthesis. The structural equation models (SEMs) revealed that plant growth, soil edaphic factors, and soil metabolites had direct or indirect influences on soil microbial communities under different organic acid conditions. Our results suggest that any phthalic or palmitic acid concentrations at concentrations >250 mg/kg are detrimental to multiple aspects of Sanqi cultivation, confirming the presence of Sanqi CCOs in Sanqi–pine agroforestry systems.

**Keywords:** *Panax notoginseng*; Sanqi–pine agroforestry systems; continuous cropping obstacles; organic acid; soil microbes; metabolites

## 1. Introduction

Continuous cropping obstacles (CCOs) refer to the abnormal crop growth and development phenomenon caused by continuous cultivation of the same crop or related crops on the same soil. During the cultivation of Sanqi, a valuable medicinal plant, CCOs present a significant challenge. One factor contributing to the formation of CCOs is the accumulation of autotoxic substances in the root exudates of Sanqi plants [1]. Previous studies have identified various types of autotoxic substances, including saponins, organic acids, flavonoids, sugars, and amino acids, in the root exudates of Sanqi cultivated using conventional management methods [2]. These substances have a negative impact on the growth and health of Sanqi. For instance, malic acid can induce bacterial wilt and infect the root of Sanqi [3], while organic acids (e.g., phthalic, stearic, palmitic, and benzoic acid) result in toxic effects

on Sanqi seedling growth [4]. Saponins (ginsenosides Rg1, Rb1; notoginseng saponins R1; and ginsenosides Rh1) can inhibit root cell viability and seedling growth, leading to root rot [5,6]. Some amino acids (lysine, isoleucine, arginine, phenylalanine, cysteine, and glycine) and sugars (D-arabinose and L-rhamnose) can promote the growth of pathogenic microbes and trigger CCOs [7], while flavonoids (quercetin) can lead to the inhibition of Sanqi seedling growth [6]. Furthermore, the soil organic acids of Sanqi cultivated using conventional management methods mainly consist of p-hydroxybenzoic, syringic, ferulic, p-coumaric, vanillic, and benzoic acid [8]. However, the organic acids in the soil of Sanqi cultivated in the agroforestry system are mainly phthalic acid and palmitic acid, according to our previous studies [9]. These results suggest that different planting patterns can affect the secretion and soil metabolites of Sanqi. Therefore, it is necessary to analyze the effects of Sanqi cultivated in the forestry system on the changes in soil metabolic substances and pathways using metabolomics technology. With the use of this technology, a judgment may be made as to whether there are CCOs.

Both mixed organic acids and single organic acids can simulate CCOs [10], which affect various aspects of plant growth, as well as edaphic factors, microorganisms, and soil metabolites. The positive or negative functions of organic acids depend on their type, concentration, and composition. For example, single organic acids have positive or negative impacts on the growth of *Panax ginseng* and *Glycine max* at low or high concentrations, respectively [11,12]. However, mixed organic acids significantly inhibit the growth and survival rate of *Panax notoginseng* and *Andrographis paniculata* [9,10]. The applied organic acids can also alter the contents of soil nutrients and organic matter. For instance, p-hydroxybenzoic acid can reduce the contents of available nitrogen (AN), available potassium (AK), and soil organic matter (SOM) in *Cunninghamia lanceolata* and *Schima superba* soils [13], while phenylalanic and p-hydroxybenzoic acid can reduce the contents of alkaline hydrolysis N, available P, available K, and SOM in the *Citrullus lanatus* soils [14]. Conversely, mixed organic acids (benzoic, p-hydroxybenzoic, ferulic, vanillin, and cinnamic acid) can decrease the contents of $NH_4^+$-N, K, and Fe in poplar soil while promoting the contents of Mn, Zn, and Cu [15]. The same types of organic acids have different impacts on bacterial and fungal diversity. For example, benzoic acid can decrease bacterial diversity but increase fungal diversity in soils growing poplar [16]. The influences of the types and concentrations of organic acids on the abundance and diversity of microbes in the same plant soil are also inconsistent. For example, fungal abundance in poplar soil first increases and then decreases with the increase in ferulic and cinnamic acid concentrations, while it increases with the increase in benzoic acid concentrations [17]. Salicylic and p-hydroxybenzoic acid at different concentrations can cause inconsistent variations in the bacterial and fungal diversity of *Vitis vinifera* rhizosphere soil [16]. Additionally, organic acids can modulate soil metabolites by interacting with other organic and phenolic acids [18]. For instance, the continuous cultivation of *Ananas comosus* for 15 years resulted in reduced sugar content (sophorose and pinotriose) and increased organic acid content (organic acid metabolites) in the soil, indicating soil quality deterioration [19]. As previously mentioned, mixed organic acids, as opposed to single organic acids, can significantly decrease the plant survival rate [12] and enhance pathogenic bacteria [9,20] while inhibiting beneficial bacteria [20] due to the stronger autotoxicity caused by the synergistic influence of mixed organic acids [21]. Most of the previous research has focused on the effects of a single organic acid on plant growth, soil physicochemical properties, and microorganisms. However, there is limited available information that compares mixed organic acids with single organic acids in the simulation of CCOs.

In our previous study, four organic acids as a mixture were applied to simulate Sanqi CCOs, and the results showed that a soil organic acid concentration of >150 mg/kg indicated the occurrence of Sanqi CCOs [9]. However, it is more challenging to determine each organic acid concentration representing Sanqi CCOs in soil. Therefore, the different concentrations of single organic acid to simulate the Sanqi CCOs are more conducive to the determination of the organic acid threshold in soil. Here, we selected exogenous

organic acids of phthalic acid, palmitic acid, and phthalic acid + palmitic mixed acid with concentrations of 0, 50, 250, and 500 mg/kg, respectively, to explore which concentration represents the occurrence of Sanqi CCOs. High-throughput sequencing and LC-MS technology were employed to elucidate the changes in soil microorganisms and metabolites, respectively. Moreover, plant growth and the physicochemical properties of the soil were measured. We hypothesized that (1) different exogenous organic acids had different effects on plant growth, edaphic factors, soil microorganisms, and soil metabolites; (2) all three types of organic acids had a common threshold, which could represent the occurrence of Sanqi CCOs.

## 2. Materials and Methods

### 2.1. Study Area

The Sanqi cultivation base is located in Xundian Hui and Yi Autonomous County, Kunming, China (103°12′ E, 25°28′ N). The site has an altitude of 2199 m, an average annual rainfall of 1900 mm, and an average annual temperature of 14.5 °C. The dominant tree species in the forests is *Pinus armandii*, with an average height of 9.5 m.

### 2.2. Experimental Design and Sample Collection

Sanqi seeds were sown into soil in a greenhouse in November 2018, with a row spacing of 5 cm × 5 cm and a depth of 1–2 cm. The soil physicochemical properties were 10–20 meq/100 g CEC, 0.2~0.4 mS/cm EC, 5.5–7.0 pH, organic matter > 2%, total nitrogen > 150 mg/kg, available phosphorus > 50 mg/kg, and available potassium > 150 mg/kg. After one year, ridging was performed under the *Pinus armandii* forest in December 2019, with a ridge height of 30–40 cm, a ridge bottom width of 120–150 cm, and a ridge top width of 80–100 cm. The Sanqi roots during the dormancy period were then transplanted to the ridges with a plant spacing×row spacing of 10 cm × 15 cm and a depth of 3–5 cm. The Sanqi was immediately mulched with 2–5 cm soil, covered with pine needles, and watered daily without fertilizer and pesticides. In our previous study, UPLC/Q-TOF-MS was performed to detect and identify the types and ratios of organic acids in root secretions and soil of Sanqi. Hence, the phthalic:palmitic acid ratio (10:1) as an optimal ratio was selected in our experiment [9]. A total of 6 g of phthalic acid, 6 g of palmitic acid, and mixed acids (6.0 g of phthalic acid + 0.6 g of palmitic acid) were dissolved in 1 mL ethanol solution (0.3%), respectively, and then diluted with water to 0.5 g/L, 25 g/L, and 5 g/L. Subsequently, different concentrations of organic acids (200 mL/plant) were applied to the roots of the Sanqi during vegetative growth phase (10 September 2021). The final organic acid concentrations of control (CK), phthalic acid (L), palmitic acid (P), and phthalic acid + palmitic acid (LP) in soils were divided into 0, 50, 250, and 500 mg/kg, respectively. A total of 60 homogeneous plots (10 m × 10 m/plot, *n* = 3) were randomly set up, including 2 sampling periods × 3 replicates = 6 CK treatments; 3 types of organic acid treatments × 3 concentrations × 2 sampling periods × 3 replicates = 54 treatments. Each plot (10 m × 10 m) included spaces (1 m) between the ridges. Five plants in each plot were harvested on the 15th day (25 September 2021) and the 30th day (10 October 2021) after organic acid treatment; the roots, stems, and leaves of the Sanqi and the soils attached to the roots with <2 mm of rhizosphere soil were sampled, respectively [22]. The Sanqi tissues and the rhizosphere soils were stored in liquid nitrogen and on dry ice, respectively, and then brought back to the laboratory and stored at −80 °C for further analysis.

### 2.3. Fresh Weight and Edaphic Factors

To determine the fresh weight of the plants, the Sanqi plants were rinsed with water and dried immediately with filter paper. The soil was sieved through a 40-mesh screen and subjected to physical and chemical analyses. Oven drying at 60 °C for 24 h was used to determine the soil water content (WC). The soil pH was measured with a pH electrode in a 5:1 water/soil mixture. The soil organic matter (SOM) was quantified using the $K_2Cr_2O_7$ oxidation method. A continuous flow analyzer (SEAL Analytical AA3) was used to measure

the ammonium nitrogen ($NH_4^+$-N), nitrate nitrogen ($NO_3^-$-N), total nitrogen (TN), and total phosphorus (TP) in the soil. The total potassium (TK) in the soil was analyzed using atomic emission spectrometry on an AA-6300C flame photometer.

### 2.4. DNA Extraction, PCR Amplification, and Sequencing

An MP-Soil (Fast DNA® Spin Kit for Soil) (MP Biomedicals, Santa Ana, CA, USA) extraction kit was used to extract the DNA of the rhizospheric soils. The quality and the concentration of the DNA were assessed using 1.0% agarose gel electrophoresis and a NanoDrop® ND-2000 spectrophotometer (Thermo Scientific Inc., Waltham, MA, USA), respectively. The 16S rRNA (V3–V4 region) and the ITS gene (ITS1 region) were amplified with primers 338F/806R for bacteria and ITS1F/ITS2R for fungus, respectively. The primers [23,24] and protocols of the PCR amplification are listed in the Supplementary Material (Table S1). The PCR products were purified with the AxyPrep DNA Gel Extraction Kit (Axygen Biosciences, Union City, CA, USA) and quantified with a Quantus™ Fluorometer (Promega, WI, USA). The purified amplicons were pooled in equimolar ratios and sequenced with paired-end reads on an Illumina MiSeq PE300 platform using the standard protocols provided by Majorbio Bio-Pharm Technology Co., Ltd. (Shanghai, China). The OTU sequence similarity threshold was 0.97. The bacterial OTUs were annotated using the databases of Silva (http://www.arb-silva.de, accessed on 15 February 2024) and Greengenes (http://greengenes.secondgenome.com, accessed on 15 February 2024), and the fungal OTUs were annotated using Unite (Release 8.0 http://unite.ut.ee/index.php, accessed on 15 February 2024). For each OTU, a representative sequence was selected, and the OTU abundance information was normalized.

### 2.5. Metabolite Profiling from Rhizosphere Samples

A total of 1 g of each rhizosphere soil from Sanqi was accurately weighed and then ground into homogenate with liquid nitrogen. The soil was mixed with 1 mL of extraction solution (methanol:water = 4:1 (*v:v*)), ground for 6 min in a frozen tissue grinder (–10 °C, 50 Hz), and subsequently stored at −20 °C for 30 min. After centrifugation for 15 min (13,000 *g*, 4 °C), 120 μL of complex solution (acetonitrile:water = 1:1) was added, and the sample was extracted using low-temperature ultrasonication for 5 min (5 °C, 40 KHz). The sample was centrifuged again for 15 min (13,000× *g*, 4 °C), and the supernatant was transferred to sample vials for LC-MS analysis using the UHPLC-Q Exactive HF-X system (Thermo Scientific, Waltham, MA, USA). The raw data were processed using Progenesis QI (Waters Corporation, Milford, MA, USA) and annotated using the Human Metabolome Database (HMDB), the Kyoto Encyclopedia of Genes and Genomes (KEGG), and other public and customized databases to identify the metabolites.

### 2.6. Statistical Analysis

Statistical Product Service Solutions (SPSS 26) software was used to conduct a one-way analysis of variance (ANOVA) test to obtain significant differences in the Sanqi fresh weight, soil physicochemical properties, and α-diversity ($p < 0.05$). The α-diversity (Chao and Shannon indices) and β-diversity were calculated using QIIME and the Bray–Curtis distance matrix, respectively [25,26]. Circos (https://cloud.oebiotech.cn/task/detail/circos/, accessed on 11 February 2024) was used to analyze the composition of microbiomes (phylum and genus). PCoA (principal coordinates analysis) was used to analyze the similarities or differences in the bacterial and fungal communities. Distance-based redundancy analysis (db-RDA) was performed using the Vegan v2.5-3 package. The 'Hmisc' (v3.5.3) and 'psych' (v2.4.1) R packages were used to calculate the correlation coefficient of the microbial abundance ($r > 0.7$, $p < 0.05$); Spearman's correlation matrices were used to filter out species in the sample with an average relative abundance of <1%. Then, we used Gephi.lnk software (v0.10.1) to draw and analyze the co-occurrence networks [27]. The bacterial and fungal stability were analyzed using R software (v4.2.2). Low AVD values indicate high stability [28].

Variance analysis was performed on the matrix file after data preprocessing. The 'ropls' R package (Version 1.6.2) was used to perform orthogonal least partial squares discriminant analysis (OPLS-DA) and to evaluate the stability of the model using 7-cycle interactive validation. A Student's *t*-test and fold difference analysis were also performed. The significantly different metabolites were selected based on the variable importance in the projection (VIP) obtained using the OPLS-DA model and the *p*-value of the Student's *t* test, with the criteria of VIP > 1 and $p < 0.05$. A total of 414 DMs were screened (https://www.kegg.jp/kegg/pathway.html, accessed on 15 February 2024) and exploited to perform metabolic enrichment and pathway analysis using scipy.stats (Python packages, v3.6.13) (https://docs.scipy.org/doc/scipy/, accessed on 15 February 2024).

Structural equation model (SEM) was implemented using SPSSPRO1.1.4 to evaluate the relationships between soil microbial α-diversity, plants, edaphic factors, and soil metabolites and to further explain the possible direct and indirect effects of the variables. Before SEM construction, pathways were constructed based on principal component analysis (PCA) to create a multivariate functional index [12]. The best-fit model was evaluated using a nonsignificant path ($p > 0.05$), and an $x^2$ test of the final model fit was performed using the model.

## 3. Results

### 3.1. Analysis of the Fresh Weight of Sanqi and Soil Physicochemical Properties under Organic Acid Conditions

Compared with the control (CK), the fresh weight of whole Sanqi plants increased at concentrations of phthalic acid and palmitic acid of <250 mg/kg but decreased at concentrations of >250 mg/kg for all the organic acids tested (Tables S2–S4). The soil water content (WC) and total phosphorus (TP) were higher, and the soil pH was lower at concentrations of <250 mg/kg than in the CK, whereas soil TN, $NH_4^+$-N, WC, TP, and pH were lower at concentrations of >250 mg/kg than in the CK. After 15 days, the soil organic matter (SOM) increased and $NH_4^+$-N decreased at concentrations of <250 mg/kg, while nitrate nitrogen ($NO_3^-$-N) decreased at concentrations of >250 mg/kg. After 30 days, $NO_3^-$-N, TN, and $NH_4^+$-N were higher, and TK and SOM were lower at concentrations of <250 mg/kg than at concentrations of >250 mg/kg (Figure S1).

### 3.2. α- and β-Diversity of Bacteria and Fungi Analysis

After operational taxonomic unit (OTU) clustering at a level of 97% similarity, 8383 bacterial OTUs and 897 fungal OTUs were obtained. Compared to the CK, the soil bacterial α-diversity increased at organic acid concentrations of >250 mg/kg, while soil fungal α-diversity increased at all concentrations, with the highest and lowest values at 250 and 500 mg/kg, respectively. No significant differences in soil bacterial α-diversity between the different organic acid concentrations or in the soil fungal α-diversity at 500 mg/kg were observed at 15 and 30 days, but the soil fungal α-diversity differed significantly at the 50 and 250 mg/kg concentrations (Figure S2).

Compared to the CK, the bacterial β-diversity was enhanced and reduced by palmitic acid concentrations of <250 mg/kg at 15 and 30 days, respectively. The lowest bacterial β-diversity was found at the mixed organic acid concentration of 500 mg/kg at 15 and 30 days. No significant difference was observed in the phthalic acid treatments. The fungal β-diversity did not change significantly in most of the treatments, except for those with the phthalic acid concentration of 500 mg/kg (15 days), the palmitic acid concentration of 500 mg/kg (30 days), and the mixed organic acid concentration of 250 mg/kg (15 and 30 days) (Figure 1).

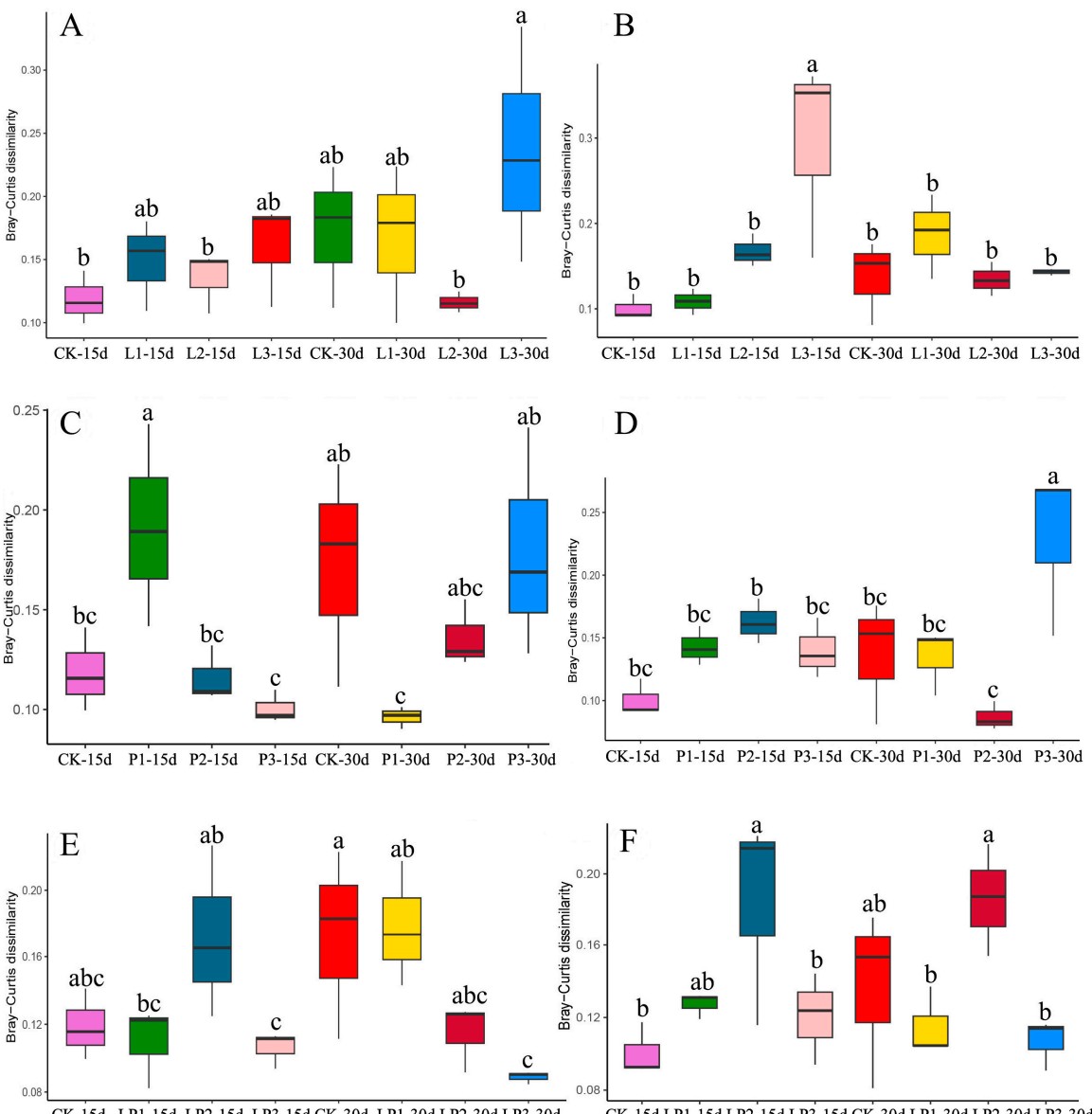

**Figure 1.** The bacterial (**A**,**C**,**E**) and fungal (**B**,**D**,**F**) β-diversity were estimated based on a Bray–Curtis distance matrix of all 60 soil samples under different organic acid treatments. (**A**–**F**) denote the treatments of phthalate, palmitic, and mixed organic acid, respectively. a, b, and c indicate significant differences at $p < 0.05$, respectively. Different colors represent different treatments.

The first and second principal components (PC1 and PC2) explained the variation in the bacterial (16.14% and 13.72%) and fungal community structure (25.98% and 15.16%), respectively. The results indicated that the type of organic acid, but not the concentration or the duration, had the greatest impact on the bacterial and fungal community structure ($R^2 = 0.115$ and 0.207, $p < 0.001$) (Figure 2).

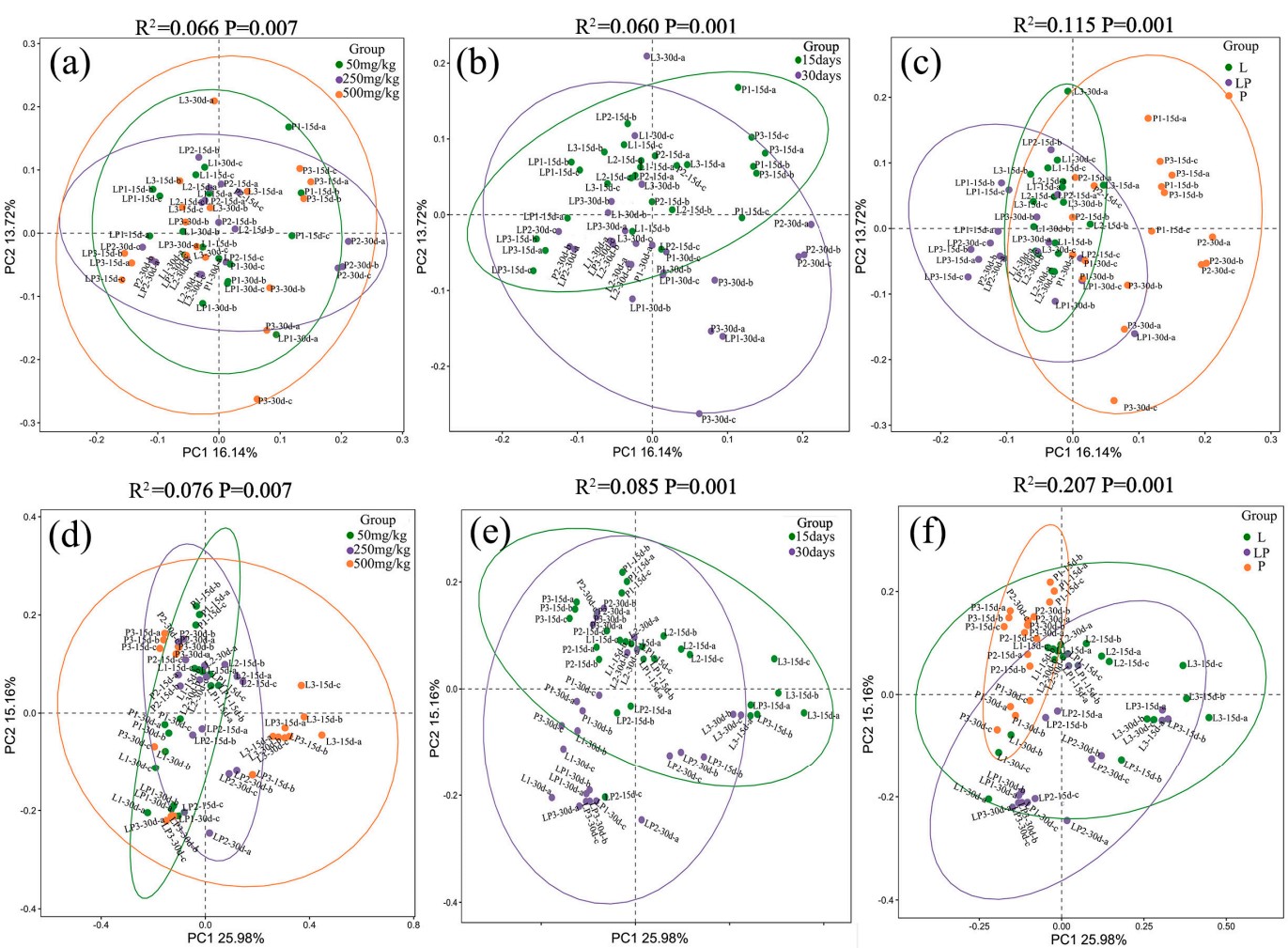

**Figure 2.** Principal coordinate analysis (PCoA) of bacterial and fungal communities. (**a**,**d**): concentrations (**b**,**e**): times (**c**,**f**): organic acid types.

### 3.3. Dominant Phyla and Genus of Bacteria and Fungi

The soil bacterial community was composed of 41 phyla, 126 classes, 303 orders, 485 families, 917 genera, and 1995 species. The most abundant phyla were Proteobacteria (36.89%), Actinobacteriota (21.95%), Acidobacteriota (15.63%), and Chloroflexi (9.65%), accounting for 84.12% of the total sequences. Compared to the CK, the abundance of Proteobacteria decreased under phthalic acid and mixed organic acid but increased under palmitic acid. The opposite trend was observed for Acidobacteriota and Chloroflexi. Actinobacteriota decreased under phthalic acid and mixed organic acid concentrations of >250 mg/kg but increased under palmitic acid (Figure 3a). The top 10 bacterial genera were *Badyrizobium* (4.18%), *Acidobacteriales* (4.0%), *Elsterales* (3.85%), *Gaiellales* (3.19%), *Burkholderia-Caballeronia-Paraburkholderia* (3.06%), *Xanthobacteriaceae* (2.77%), *Acidothermus* (2.76%), *IMCC26256* (2.46%), *Subgroup_2* (2.39%), and *Vicinamidobacteriales* (2.34%) (Figure S3). All three organic acids decreased the abundances of *Badyrhizobium*, *Elsterales*, and *Xanthobacteriaceae*, whereas they increased the abundances of *Gaiellales*, *Burkholderia-Caballeronia-Paraburkholderia*, and *Acidothermus*, compared to the CK. *Acidobacteriales* increased under phthalic acid and mixed organic acid but decreased under palmitic acid (Figure 3b).

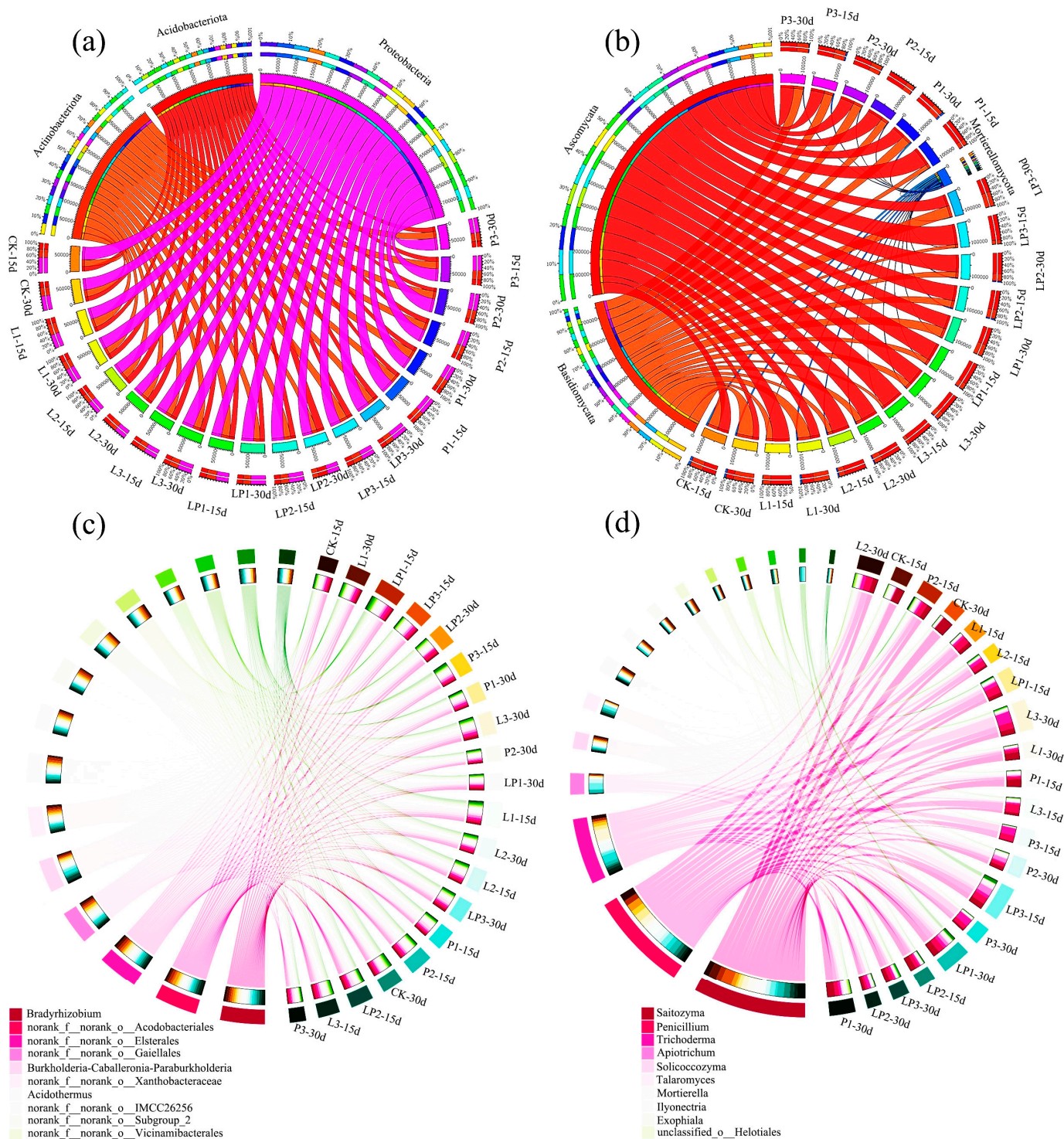

**Figure 3.** The relative abundance of the bacteria (**a**,**c**) and fungi (**b**,**d**). (**a**,**b**): phylum and (**c**,**d**): genus.

The fungal community consisted of 9 phyla, 37 classes, 86 orders, 164 families, 297 genera, and 429 species. The dominant phyla were Ascomycota (62.97%), Basidiomycota (32.84%), and Mortierellomycota (3.33%), which represented 99.14% of the total species. Compared to the CK, all the organic acid types increased the abundance of Ascomycota but decreased the abundance of Basidiomycota and Mortierellomycota (Figure 3c). The top 10 fungal genera were *Saitozyma* (23.48%), *Penicillium* (20.29%), *Trichoderma* (13.58%), *Apiotrichum* (3.6%), *Solicocozyme* (3.34%), *Talaromyces* (3.23%), *Mortierella* (3.23%), *Ilyonectria* (2.41%), *unclassified_o__Helotiales* (2.13%), and *Exophiala* (2.1%), accounting for 77.39% of the

total sequences (Figure S4). All three organic acids reduced the abundances of *Saitozyma*, *Mortierella*, *Ilyonectria*, *unclassified_o__Helotiales*, and *Exophiala* but increased the abundances of *Penicillium*, *Apiotrichum*, *Trichoderma*, *Talaromyces*, and *Fusarium*, compared to the CK (Figure 3d).

SOM, pH, TN, WC, and TK were the key factors influencing the bacterial community structure. NH$_4^+$-N, pH, TK, and TP were the main factors affecting the fungal community structure (Figure 4 and Table S5).

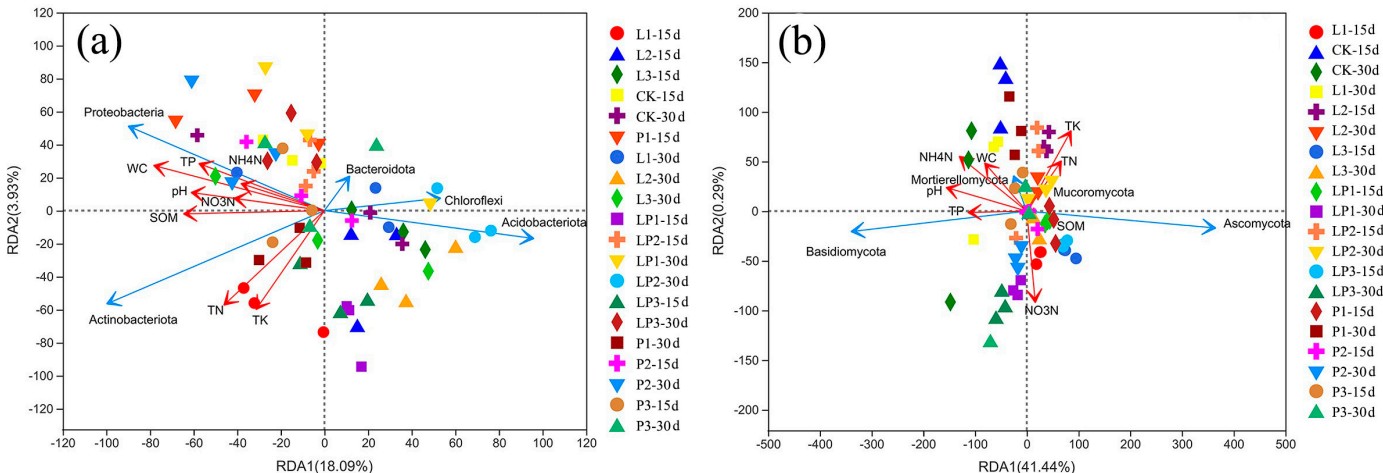

**Figure 4.** The RDA analysis plots of the (**a**) bacteria/(**b**) fungi and edaphic factors.

### 3.4. Complexity and Stability of Bacterial and Fungal Network

The influence of the organic acid concentrations of >250 mg/kg on the microbial communities was evaluated by conducting microbial network complexity and community stability analyses (Figures S5 and S6; Tables S6 and S7). The bacterial and fungal networks had fewer nodes and edges and lower average degrees, graph densities, and average clustering coefficients but larger network diameters and average path lengths under the different organic acid concentrations and types compared to the CK. The microbial network complexity followed the order of CK > 50 > 250 and 500 mg/kg (Figure 5). The average variation degree (AVD) of the bacterial community increased significantly under the mixed organic acids but not under the single organic acids, indicating a lower stability of the bacterial community. The fungal community stability was also reduced by the organic acid concentrations of >250 mg/kg, as indicated by the higher AVD value after 30 days, but not after 15 days of treatment (Table 1).

**Table 1.** Average variation degree of the microbial communities under organic acids.

| | AVD of Bacterial Community (15 Days) | AVD of Bacterial Community (30 Days) | AVD of Fungal Community (15 Days) | AVD of Fungal Community (30 Days) |
|---|---|---|---|---|
| CK | 0.66 ± 0.09 [bc] | 0.651 ± 0.01 [b] | 0.88 ± 0.05 [a] | 0.73 ± 0.03 [ab] |
| L (<250 mg/kg) | 0.78 ± 0.14 [ab] | 0.666 ± 0.19 [b] | 0.65 ± 0.02 [b] | 0.60 ± 0.09 [c] |
| L (>250 mg/kg) | 0.62 ± 0.06 [c] | 0.818 ± 0.12 [abc] | 0.60 ± 0.03 [b] | 0.80 ± 0.03 [a] |
| LP (<250 mg/kg) | 0.79 ± 0.04 [ab] | 0.693 ± 0.06 [bc] | 0.68 ± 0.05 [b] | 0.69 ± 0.05 [b] |
| LP (>250 mg/kg) | 0.82 ± 0.15 [a] | 0.875 ± 0.0 [a] | 0.66 ± 0.06 [b] | 0.80 ± 0.02 [a] |
| P (<250 mg/kg) | 0.76 ± 0.04 [abc] | 0.686 ± 0.0 [bc] | 0.59 ± 0.05 [b] | 0.73 ± 0.05 [ab] |
| P (>250 mg/kg) | 0.73 ± 0.12 [abc] | 0.848 ± 0.06 [ab] | 0.64 ± 0.1 [b] | 0.78 ± 0.01 [a] |

Note: a, b, and c indicate significant differences at *p* < 0.05, respectively.

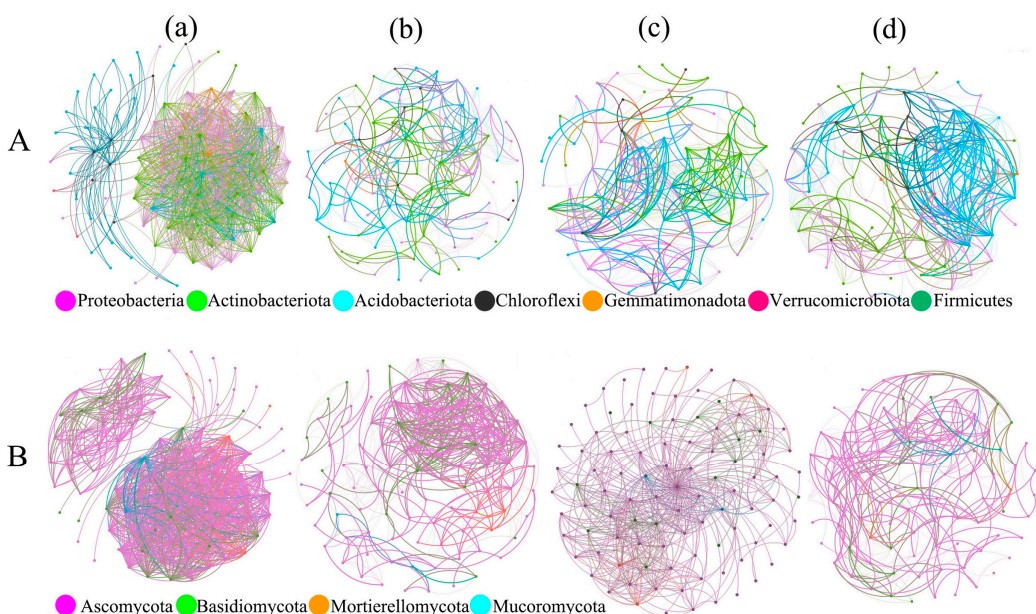

**Figure 5.** Variation of the complexity of bacterial and fungal networks at organic acid concentrations >250 mg/kg. (**A**,**B**) indicate bacterial and fungal networks. (**a**–**d**): CK, phthalic acid, palmitic acid, and mixed organic acids, respectively.

### 3.5. Differential Metabolites (DMs) in Sanqi Soils

The LC-MS analysis detected 588 metabolites in the rhizosphere soils of Sanqi; the metabolites belonged to various classes, such as lipids (22.4%), terpenoids (21.6%), carbohydrates (7.7%), organic heterocyclic compounds (5.1%), amino acids and derivatives (4.6%), flavonoids (3.1%), benzenoids (3.1%), nucleotides and derivatives (2.7%), phenylpropane and polyketide compounds (2.4%), organic acids (2.0%), coumarins (1.9%), phenolic acids (1.5%), organic oxygen compounds (1.5%), organic nitrogen compounds (1.0%), alcohols and polyols (0.9%), lignans (0.2%), and others (18.4%).

The phthalic acid treatment resulted in 138 differentially abundant metabolites (DMs). The 45 up-regulated metabolites included benzenoids, amino acids and derivatives, and phenolic acids, while the 93 down-regulated metabolites included organic heterocyclic compounds, terpenoids, carbohydrates, and nucleotides and derivatives. The mixed organic acid treatment led to 159 DMs. The 64 up-regulated metabolites included benzenoids, phenolic acids, and ginsenosides, while the 95 down-regulated metabolites included organic heterocyclic compounds, nucleotides and derivatives, flavonoids, and coumarins. The palmitic acid treatment resulted in 117 DMs. The 39 up-regulated metabolites included benzenoids and phenylpropane and polyketide compounds, while the 78 down-regulated metabolites included organic oxygen compounds, organic acids, nucleotides and derivatives, and coumarins (Figure S7).

The top 30 DMs under the 3 organic acid treatments were analyzed using VIP. The phthalic acid treatment enriched many compounds in the classes of lipids (Araliacerebroside), oxygenated organic compounds (DHAP (10:0)), ginsenosides ((20R)-Ginsenoside Rh2), and organic heterocyclic compounds (Doxepin N-oxide glucuronide, Cucurbitacin I 2-glucoside). The palmitic acid treatment enriched phenolic acid (5-(3,4,5-trihydroxyphenyl) pentanoic acid), organic oxygen compounds (4-hydroxy-2,6,6-trimethyl-3-oxo-1,4-cyclohexadiene-1-carboxaldehyde), lipids (Araliacerebroside), and benzenoids (Hydroxybuprenorphi). The mixed organic acid treatment enriched many compounds in the classes of terpenoids (7,8-dehydroastaxanthianthin), phenylpropane and polyketone compounds, organic oxygen compounds, phenolic acids (5-(3,4,5-trihydroxyphenyl) pentanoic acid), and ginsenosides ((20R)-ginsenoside Rh2, ginsenoside Rh6) ($p < 0.01$) (Figure S8).

The KEGG pathway database analysis revealed 17 metabolic pathways of DMs under the different organic acid treatments (Figure 6). The phthalic acid, palmitic acid, and mixed

organic acid treatments involved two (steroid hormone and flavonoid biosynthesis), three (steroid hormone, flavonoid, ubiquinone, and other terpenoid–quinone biosynthesis), and five (steroid hormone, linoleic acid metabolism, primary bile acid, ketone bodies, plant hormones, amino sugar, and nucleotide sugar) metabolic pathways, respectively, with steroid hormone biosynthesis being common to all treatments (Figure 6C and Table S8). Compared to the CK, the concentrations of Pinostrobin (L, P), 17alpha,21-Dihydroxypregnenolone, cortisol, 18-Hydroxycorticosterone (P), 3-Hexaprenyl-4,5-Dihydroxybenzoic acid, N-Acetylmannosamine, N-Acetylglucosamine, L-malic acid, and shikimic acid (LP) increased with the increasing concentration, while the concentrations of corticosterone, 11-Dehydrocorticosterone, cortisol, eriodictyol, 18-Hydroxycorticosterone(L), 3-Hexaprenyl-4,5-Dihydroxybenzoic acid (P), 13(S)-HpODE, corticosterone, taurine, and 11-Dehydrocorticosterone (LP) decreased at different concentrations (Figure 6A,B).

*3.6. Correlation of Soil Microbes with DMs*

The association between the top 15 bacterial and fungal genera and their DMs was examined to elucidate the effect of organic acid concentrations of >250 mg/kg on Sanqi growth. The results indicated that the correlation between the fungi and the metabolites was stronger than that between the bacteria and the metabolites (Figure S9). For phthalic acid, dehydroepiandrosterone sulfonate and 11 Dehydrococcosterone were correlated with *AD3* and *unclassified_f__Xanthobacteriaceae*, and they also showed a positive or negative correlation with *Mortierella*, *Ilyonectria*, and *Talaromyces*, respectively. Pinostrobin had negative and positive correlations with *unclassified_o_Helotiales* and *Trichoderma*, respectively (Figure S9a,b). For palmitic acid, 11b-Hydroxyprogesterone had a significant positive correlation with bacterial genera (*Acidobacteriales*, *Subgroup_2*, and *Micrococcaceae*) and a negative correlation with fungal genera (*Apiotrichum* and *Talaromyces*). Pinostrobin had a positive correlation with bacterial genera (*IMCC26256* and *Xanthobacteraceae*) and fungal genera (*Apiotrichum* and *Talaromyces*). 3-hexaprenyl-4,5-Dihydroxybenzoic acid had a positive correlation with *AD3* and a negative correlation with fungal genera (*Trichoderma*, *unclassified_o_Helotiales*, and *Trichophaea*) (Figure S9c,d). For the mixed organic acids, taurine had positive correlation with bacterial genera (*Burkholderia-Caballeronia-Paraburkholderia*) and fungal genera (*Saitozyma* and *Cladophialophora*) and a negative correlation with bacterial genera (*Xanthobacteraceae*, *Acidobacteriales*, and *Vicinamibacterales*) and fungal genera (*Talaromyces* and *Trichoderma*). 18-Hydroxycorticosterone had a negative correlation with bacterial genera (*Xanthobacteraceae*, *Acidobacteriales*, and *Subgroup_2*) and a negative or positive correlation with fungal genera (*Talaromyces*, *Trichoderma*, and *Saitozyma*), respectively. 13(S)-HpODE had a negative correlation with bacterial genera (*Acidobacteriales*, *Vicinamibacterales*, *Subgroup_2*, and *Candidatus_Solibacter*) and fungal genera (*Trichoderma*, *Trichophaea*, and *Pseudogymnoascus*) and a positive correlation with bacterial genera (*Gaiellales*, *unclassified_f_Xanthobacteraceae*, *Micrococcaceae*, and *Bradyrhizobium*) and fungal genera (*Apiotrichum* and *Solicoccozyma*). N-Acetylglucosamine and shikimic acid had a significant correlation with the corresponding bacteria (*Candidatus_Solibacter*, *Micrococcaceae*, and *Bradyrhizobium*) and the corresponding fungi (*Apiotrichum*, *Solicoccozyma*, *Trichophaea*, and *Pseudogymnoascus*). 3-Hydroxybutyric acid had a negative or positive correlation with fungal genera (*Mortierella*, *Talaromyces*, and *Ilyonectria* Acetylmannosamine) and had a positive correlation with the fungus *Penicillium*. 11-Dehydrocorticosterone had a negative correlation with the fungus *Fusarium*. Cortisol had a positive or negative correlation with the fungi *Apiotrichum* and *Trichophaea*, respectively (Figure S9e,f).

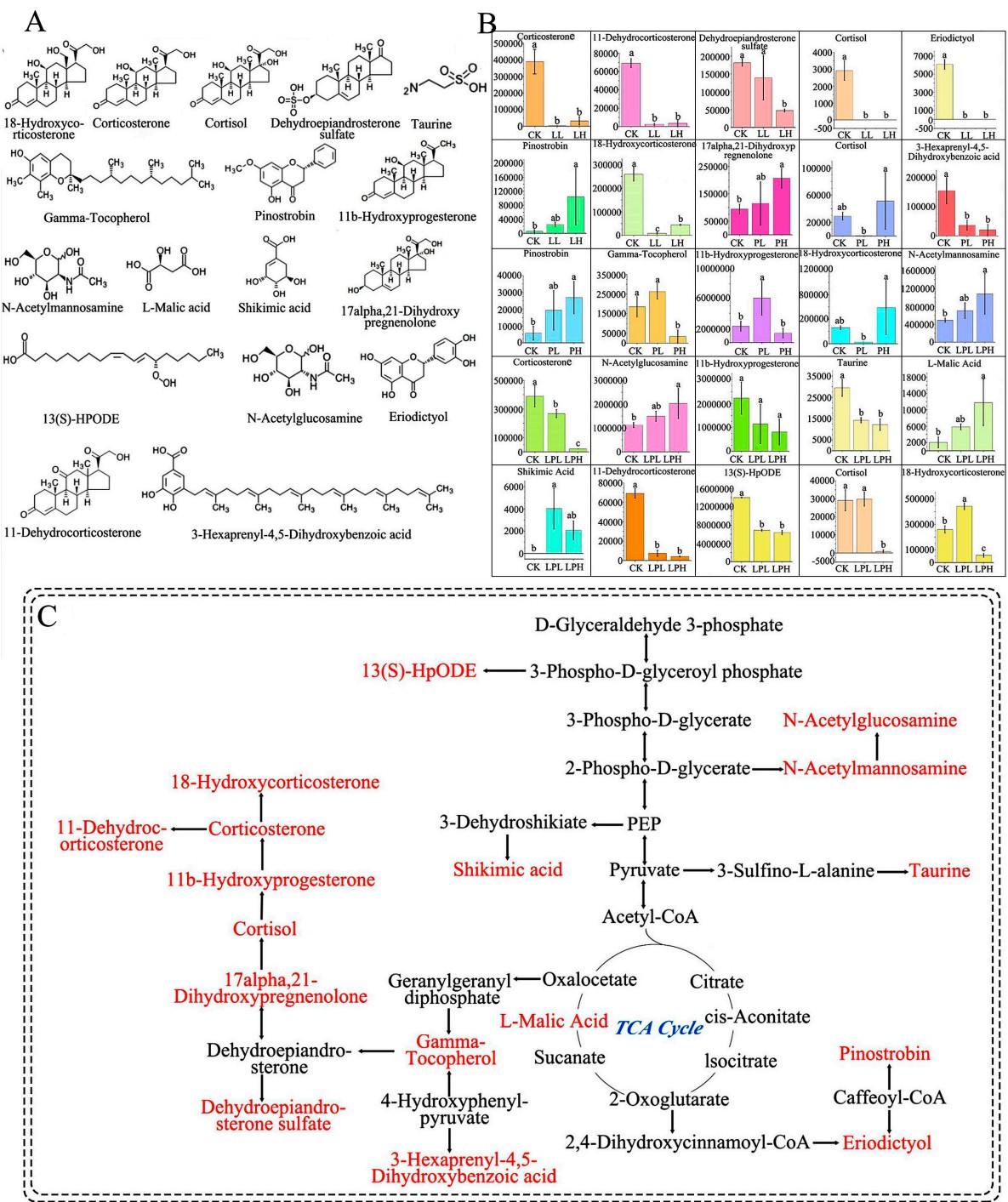

**Figure 6.** The structural formula (**A**), variation of abundance (**B**), and metabolic pathway (**C**) for 17 differential metabolites (DMs) in Sanqi soils under organic acids. LL, PL, and LPL: phthalic, palmitic, and mixed organic acid <250 mg/kg; LH, PH, and LPH: phthalic, palmitic, and mixed organic acid >250 mg/kg. a, b, and c indicate significant differences at *p* < 0.05, respectively.

### 3.7. Structural Equation Model (SEM) Analysis

The direct factors influencing microbial diversity under the organic acid concentrations of >250 mg/kg were analyzed using SEM (Figure 7 and Tables S9–S12). In the CK treatment, the plants had a significant impact on bacterial and fungal α-diversity, while the metabolic substances had a significant impact on fungal α-diversity (Figure 7a). The plants, metabolic substances, and physicochemical properties had a significant effect on fungal α-diversity

under the phthalic acid treatment (Figure 7b). The physicochemical properties had a significant impact on bacterial $\alpha$-diversity under the palmitic acid treatment (Figure 7c). The physicochemical properties had a significant impact on fungal $\alpha$-diversity under the mixed organic acid treatment (Figure 7d).

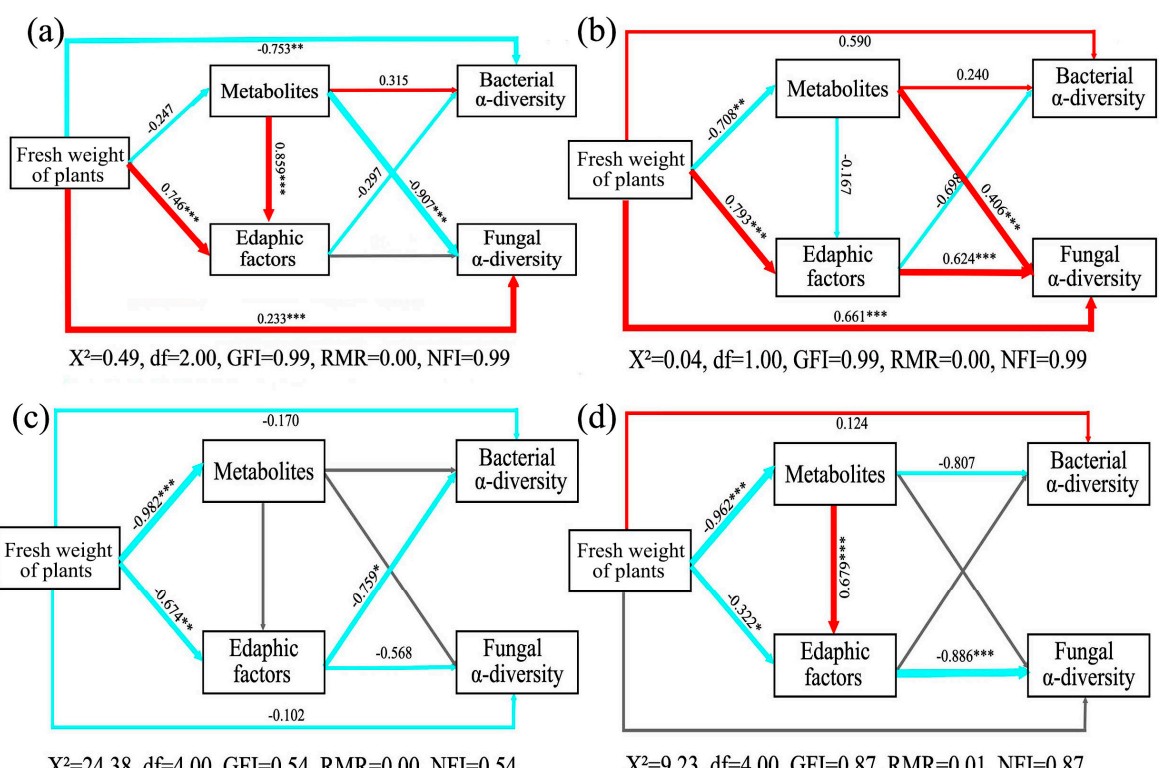

**Figure 7.** Structural equation model (SEM) analysis for organic acids >250 mg/kg. The width and the number on arrow indicate the path strength and the effective coefficient, respectively. The blue or red arrows represent positive or negative causality, respectively. (**a**): CK; (**b–d**): phthalic, palmitic, and mixed organic acid. * $p < 0.05$, ** $p < 0.01$, *** $p < 0.001$.

## 4. Discussion

### 4.1. Organic Acid Concentrations of >250 mg/kg Reduced Sanqi Growth and Soil's Physicochemical Properties

The soil's physicochemical properties were affected by the different types, concentrations, and durations of organic acid treatments [15]. We found that the contents of WC and TP increased and the pH value decreased at concentrations of <250 mg/kg, while the contents of TN, $NH_4^+$-N, WC, and TP and the pH values decreased at concentrations of >250 mg/kg. These results could be explained by the fact that the concentration of organic acids can modulate the degree of facilitatory or inhibitory effects of environmental factors [29]. Furthermore, organic acid treatments could also increase or decrease the content of some soil nutrients over time, as has been found in previous research [15].

Organic acid concentrations also had different effects on the growth and fresh weight of plants [1]. In our study, the fresh weight of the Sanqi decreased at concentrations of >250 mg/kg, in agreement with the previous results [30]. Moreover, plant growth was significantly inhibited by the mixed organic acid treatments [9,31,32]. Interestingly, the mixed organic acids had a stronger inhibitory effect on Sanqi growth than the single organic acids, which might be due to the inhibition of the photosynthetic rate and the substance synthesis by mixed organic acids [2]. Moreover, the plant growth was significantly positively correlated with WC, TP, SOM, $NH_4^+$-N, TN, and TK but negatively correlated with $NO_3^-$-N and not correlated with pH (Table S13). We hypothesized that organic acids could affect the soil physicochemical properties or the root absorption function, as well

as the differential nutrient preferences of the roots, and consequently affect plant growth. Meanwhile, organic acids significantly affected the association of the root of Sanqi with interactive microbes, further altering soil nutrition.

### 4.2. Organic Acid Concentrations of >250 mg/kg Increased Microbial α-Diversity

Both bacterial and fungal α-diversity increased under organic acid concentrations of >250 mg/kg. Previous results showed that fungal α-diversity, but not bacterial α-diversity, increased under mixed organic acid concentrations of >150 mg/kg in Sanqi soils [12]. This could be attributed to the types of exogenous organic acids, as mixed organic acids (synergistic effects) could interact with other metabolites and organic acids and alter the bacterial and fungal communities. Moreover, exogenous organic acids at different concentrations could alter the abundance and α-diversity of bacteria and fungi [16,17]. The bacterial and fungal diversity of the Sanqi soil responded differently to the organic acid types, which was consistent with the results of previous studies [16]. The correlation analysis revealed that the bacterial diversity was significantly correlated with WC, pH, TP, $NH_4^+$-N, and $NO_3^-$-N, while the fungal diversity was correlated with TK (Table S14). We hypothesized that the organic acids changed the soil's physicochemical properties and consequently affected the bacteria and fungi. Furthermore, increased fungal diversity and decreased bacterial diversity were major factors associated with Sanqi root rot [33]. Notably, all three types of organic acids with different concentrations and durations increased the fungal diversity to varying extents, especially the root rot pathogens.

### 4.3. Different Organic Acids Affect the Dominant Flora in Sanqi Soils Differently

At the phylum level, the abundance of dominant bacterial groups changed inconsistently under the different organic acid treatments, following the order of Proteobacteria > Actinobacteria > Acidobacteria. Proteobacteria and Actinobacteriota mainly participate in the decomposition of organic matter and C/N cycling [3,23], respectively. Acidobacteriota are acidophilic and can influence the soil pH value [34]. The abundance of dominant fungal groups changed consistently under the different organic acid treatments, following the order of Ascomycota > Basidiomycota > Mortierellomycota. Ascomycota are mainly related to the decomposition of litter or residues in the ecosystem [35]. Basidiomycota are mainly related to the decomposition of lignin and cellulose [36]. Mortierellomycota have various functions, such as the decomposition of organic matter, the promotion of plant root absorption, and the inhibition of pathogens [37]. RDA indicated that SOM, pH, TN, WC, and TK were the key factors that significantly influenced the dominant bacterial community and that $NH_4^+$-N, pH, TK, and TP were the key factors that influenced the fungal community. There was a cover formed by a large number of pine needles during the planting of Sanqi, which may have enhanced the soil nutrient content and increased the microbial abundance [38]. Therefore, the application of exogenous organic acids might increase the bacterial and fungal flora involved in the nutrient cycle and the litter decomposition, respectively. Three organic acids decreased the amount of beneficial (*Badyrhizobium*) and pathogenic bacteria (*Xanthobacteriaceae, Ilyonectria*) but increased the amount of other pathogenic (*Fusarium*) and harmful microbes (*Penicillium*); the latter bacteria have nitrogen fixation as a main function [39], which causes plant leaf spot disease [12], induces Sanqi root rot [33], causes plant wilt [40], and generates harmful compounds (mycotoxins) [41], respectively.

### 4.4. Concentration of Organic Acids of >250 mg/kg Reduced the Complexity and Stability of Microbial Network

The complexity of microbial networks reflects the nutrient availability and diversity of soil microorganisms [42]. Lower network complexity indicates greater nutrient limitation, lower soil nutrient content, and poorer plant growth [43]. The stability of microbial networks depends on the composition (abundance and species) and interaction of the soil microorganisms [44]. Lower network complexity implies lower community stability and

higher susceptibility to environmental disturbances. We found that organic acid concentrations of >250 mg/kg reduced the network complexity and stability of both the bacteria and the fungi in Sanqi soils, which was in agreement with the previous findings that showed that mixed organic acids reduced the bacterial and fungal network complexity and the fresh weight of Sanqi [12]. Similarly, *Glycine max* reduced the network stability of bacterial communities in continuously cropped soybean soils [45]. Therefore, we deduced that the reduction in soil nutrients and the inhibition of Sanqi growth by exogenous organic acids of >250 mg/kg might be the reasons for the decrease in microbial complexity and stability.

*4.5. Different Types of Organic Acids Changed Soil DMs and Metabolic Pathways*

The soil DMs and up-regulated species changed under the different organic acid treatments, following the order of mixed organic acids (159, 64) > phthalic acid (138, 45) > palmitic acid (117, 39). The mixed organic acids significantly increased the soil DMs and up-regulated species compared to the single organic acids. Previous studies reported that Ananas comosus soil had 120 and 80 DMs after 5 and 15 years of CCOs, respectively [46]. However, the number of DMs produced by the CCOs of Sanqi was much higher than the number produced by other plants, which might be due to the fact that Sanqi could release more metabolites into the soil and the fact that these metabolites could interact with each other to produce synergistic effects [18].

All three organic acids increased the content of phenolic acids (terephthalic acid and 5-(3,4,5-tetrahydroxyphenyl) pentanoic acid), and the mixed organic acids also increased the content of saponins ((20R)-Ginsenoside Rh2, Ginsenoside Rg5, Ginsenoside M7cd, Ginsenoside Rh6, Ginsenoside Rg5, and Ginsenoside Rf). These results were similar to those of previous research, which showed that phenolic acid metabolites ((2R,3S)-2-hydroxy-3-isopropylsuccinic acid, 4-hydroxybenzoic acid, 4-hydroxy-3-methoxybenzoic acid, and 3,4-dihydroxybenzoic acid) increased the CCOs in *Ananas comosus* soil and were the main cause of the CCOs [46]. We concluded that the increase in phenolic acids and saponins in Sanqi soil would lead to CCOs, as the organic acids released from Sanqi could inhibit its growth and cause changes in the rhizosphere microbiome.

The effects of the organic acid concentrations of >250 mg/kg on soil metabolic pathways showed that the 3 organic acids altered 10 metabolic pathways, all of which involved steroid hormone biosynthesis. Steroid hormone biosynthesis could reduce the metabolic activity of microbes and pollute the soil environment [47]. It is noteworthy that the abundance of 17alpha, 21-Dihydroxypregnenolone, cortisol, 18-Hydroxycorticosterone, corticosterone, and 11-Dehydrocorticosterone changed significantly under the different organic acid treatments; thus, they may be the key metabolic substances produced by the CCOs of Sanqi. Moreover, the mixed organic acids also increased the content of L-malic and shikimic acid in the biosynthesis of the plant hormones pathway. Previous results have shown that L-malic and shikimic acid could promote the growth of pathogens and the formation of phenolic acid autotoxic substances [3,48,49], respectively. Therefore, the mixed organic acids increased the production of harmful metabolites in Sanqi soil, resulting in the autotoxicity of the mixed organic acids being stronger than that of the single organic acids.

*4.6. Organic Acid Changed the Relationship among SANQI, Edaphic Factors, Metabolites, and Soil Microbes*

The bacterial and fungal diversity of Sanqi soils is affected by plants, edaphic factors, and soil metabolites, either directly or indirectly. However, the mechanisms vary depending on the types of organic acids. Previous studies have revealed a complex relationship between the microbial community structure and edaphic factors, the plant and organic acid species, and the inter-root metabolites [50–52]. Our results confirmed this finding by showing that the fresh weight of Sanqi plants had a direct effect on microbial diversity (CK), while edaphic factors had a direct effect on fungal diversity (L, P, and LP). Plant roots can supply carbon and energy sources for microbial growth and development, resulting in different impacts on microbial diversity and distribution [53,54]. Sanqi, as a medicinal plant,

can produce some specific root exudates (ginsenosides and phenolic acids) that affect microbial community composition. Moreover, rhizosphere metabolic activity can change soil chemical properties, creating a niche for specific rhizosphere microbial communities [55], while soil microbial communities can adapt dynamically to the soil environment [56].

## 5. Conclusions

Our results indicate that organic acid concentrations of >250 mg/kg reduced the fresh weight of Sanqi, the contents of TN, $NH_4^+$-N, WC, TP, and the pH value. Although the soil bacterial and fungal α-diversity increased, the complexity and stability of the microbial network decreased at concentrations of >250 mg/kg. Moreover, three types of organic acids increased the number of pathogenic *Fusarium* and harmful *Penicillium*. Furthermore, 17 DMs and 10 metabolic pathways were significantly altered under organic acid concentrations of >250 mg/kg. SEM analysis revealed that organic acid-simulated CCOs changed the relationship between the plants, soil physicochemical properties, metabolites, and soil microbes. In summary, organic acid concentrations of >250 mg/kg had adverse effects on Sanqi growth, soil microorganisms, and soil metabolism; these effects could lead to the CCOs of Sanqi.

**Supplementary Materials:** The following supporting information can be downloaded at: https://www.mdpi.com/article/10.3390/agronomy14030601/s1. Table S1: The primers and protocols of PCR amplification; Table S2: Fresh weight of plants and soil physicochemical properties under phthalic acid; Table S3: Fresh weight of plants and soil physicochemical properties under palmitic acid; Table S4: Fresh weight of plants and soil physicochemical properties under mixed organic acid; Table S5: Correlation between microbial (Bacterial, fungal) community and edaphic factors; Table S6: Different coefficients of bacterial network complexity; Table S7: Different coefficients of fungal network complexity; Table S8: Metabolic pathway description of differential metabolites under different organic acids; Table S9: The path analysis data for CK; Table S10: The path analysis data for phthalic acid conditions; Table S11: The path analysis data for palmitic acid conditions; Table S12: The path analysis data for mixed organic acid conditions; Table S13: Correlation between fresh weight and physicochemical properties of plants under different organic acid conditions; Table S14: Correlation between microbial (bacterial, fungal) diversity index (Shannon) and soil physicochemical properties. Figure S1: Soil physicochemical properties under different organic acids treatments; Figure S2: Shannon index and chao index of bacteria and fungi; Figure S3: The relative abundance of the bacterial (a, c, e) and fungal (b, d, f) communities at the phylum level; Figure S4: The relative abundance of the bacterial (a, c, e) and fungal (b, d, f) communities at the genus level; Figure S5: Changes in the complexity of bacterial networks under different organic acids; Figure S6: Changes in the complexity of fungal networks under different species organic acids; Figure S7: Volcano plot showing the different metabolites with different organic acid treatments ($p < 0.05$, VIP > 1); Figure S8: VIP values in the OPLS-DA model (VIP values > 1.0) showing DMs under the three organic acid treatments; Figure S9: Network analysis of the top 15 bacteria, fungi and DMs.

**Author Contributions:** Conceptualization, J.H. and S.W.; methodology, J.H. and S.W.; software, J.H., Y.L. and Q.W.; validation, J.H., Y.L. and Q.W.; formal analysis, J.H. and S.W.; investigation, J.H., S.W. and X.H.; resources, J.H., Y.L. and Q.W.; data curation, S.W. and X.H.; writing—original draft preparation, J.H., Y.L. and Q.W.; writing—review and editing, J.H., S.W. and X.H.; visualization, J.H., Y.L., Q.W., X.H. and S.W.; supervision, J.H. and S.W.; project administration, S.W. and X.H.; funding acquisition, J.H. and S.W. All authors have read and agreed to the published version of the manuscript.

**Funding:** This work was supported by the Yunnan Ten Thousand People Plan Youth Top Talent Project (YNWR-QNBJ-2019–028), the China Agriculture Research System (CARS-21), the Major Science and Technology Project of Yunnan Province (202102AE090042, 202202AE090036, 2019ZG0901, 2021Y250), the Kunming Science and Technology Bureau (2021JH002), and the Forestry Innovation Programs of Southwest Forestry University (Grant No. LXXK-2023M17).

**Data Availability Statement:** Data are presented in the paper. The sequence data have been deposited in the National Center for Biotechnology Information Sequence Read Archive under Accession No. PRJNA860751 (16S rRNA data) and No. PRJNA861133 (ITS data).

**Conflicts of Interest:** The authors declare no conflicts of interest.

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
