# Peer review of "Effects of Exogenous Organic Acids on the Soil Metabolites and Microbial Communities of Panax notoginseng from the Forest Understory"

_agronomy, doi:10.3390/agronomy14030601_

Round 1

Reviewer 1 Report

Comments and Suggestions for Authors

In the article, the authors studied the effect of exogenous organic acids on soil metabolites and microbial communities of Panax notoginseng from forest undergrowth. This topic is relevant because the existence of Sanqi CCOs in Sanqi–pine agroforestry systems remains unclear.

The work is relevant and highly novel. The data is well processed statistically, but there are a few questions and comments:

1. When studying control, the individual effect of 2 acids and their mixtures, the authors dissolved the acids in 0.3% ethanol. Have any experiments been carried out to study the influence of the solvent itself?

2. In lines 121-123, the authors provide a difficult-to-read processing scheme. Is it possible to make it more understandable for the reader?

3. The authors studied the effect of 0, 50, 250 and 500 mg/kg acids. In the case of a mixture of acids, the concentrations reported are the sum of the two acids or the concentrations of each of the acids being studied in the mixture (e.g. 250 mg/kg = 250 mg/kg L + 250 mg/kg P or 250 mg/kg = 125 mg/kg L + 125 mg/kg P)?

4. Why was the concentration of 250 mg/kg chosen for the main analysis?

Notes:

1. Line 148 – a link to table S1 is provided, there is no table itself.

2. Line 155 – is there a typo in the database name?

3. Line 166 - manufacturer UHPLC‒Q Exactive HF‒X system is not specified

4. Line 172 - is the full name of the SPSS26 software specified?

5. Line 207 - tables S2-S4 are not in the materials.

6. Line 214 - there is no figure S2 in the materials

7. Figure 2 - unreadable captions on the figures

8. Figure 3 - unreadable symbols

9. Line 276 – a link to table S5 is provided, which is not in the materials

10. Line 282 – references are provided to Figures S5, S6 and tables S6, S7, which are not in the materials

11. Line 297 - you should select one section name

12. Line 315 – a link is provided to Figures S7, which is not in the materials

13. Line 326-327 – a link to table S8 is provided, which is not in the materials

14. Unreadable captions in Figure 6b

15. Lines 351, 356, 362, 380-381 – references are given to Figures S9a-f, which are not in the materials

16. Line 416 – a link to table S13 is provided, there is no table itself

17. Line 430 – a link to table S14 is provided, there is no table itself

18. Line 529 - Can Fusarium and Penicillium be classified as bacteria? Is there a typo here?

Best regards, reviewer.

Author Response

Dear reviewer, Thank you for your suggestions.

1). When studying control, the individual effect of 2 acids and their mixtures, the authors dissolved the acids in 0.3% ethanol. Have any experiments been carried out to study the influence of the solvent itself?

We have improved it. Organic acids are less soluble in water, so they must be dissolved with ethanol first. Organic acids were dissolved in 1ml ethanol solution (0.3%) and then diluted with water to 0.5 g/L, 25 g/L, and 5 g/L. This ethanol concentration does not refer to the ethanol concentration in the final solution.

2). In lines 121-123, the authors provide a difficult-to-read processing scheme. Is it possible to make it more understandable for the reader?

We have improved it in line 146.

3) The authors studied the effect of 0, 50, 250 and 500 mg/kg acids. In the case of a mixture of acids, the concentrations reported are the sum of the two acids or the concentrations of each of the acids being studied in the mixture (e.g. 250 mg/kg = 250 mg/kg L + 250 mg/kg P or 250 mg/kg = 125 mg/kg L + 125 mg/kg P)?

We have improved it in line 135~144.

4) Why was the concentration of 250 mg/kg chosen for the main analysis?

We focused on the organic acid >250 mg/kg, due to these organic acid concentrations resulted in a decrease in the fresh weight of Sanqi. The results indicated the occurrence of Sanqi CCOs.

Notes:

  1. Line 148 – a link to table S1 is provided, there is no table itself.

Sorry about this problem. Due to a system error, the attached materials are not displayed. We have resubmitted the supplementary materials.

  1. Line 155 – is there a typo in the database name?

It has been corrected.

  1. Line 166 - manufacturer UHPLC‒Q Exactive HF‒X system is not specified

We have added it.

  1. Line 172 - is the full name of the SPSS26 software specified?

We have added it.

  1. Line 207 - tables S2-S4 are not in the materials.

Supplementary materials have been re-uploaded.

  1. Line 214 - there is no figure S2 in the materials

Supplementary materials have been re-uploaded.

  1. Figure 2 - unreadable captions on the figures

We have improved it.

  1. Figure 3 - unreadable symbols

We have improved it.

  1. Line 276 – a link to table S5 is provided, which is not in the materials

Supplementary materials have been re-uploaded.

  1. Line 282 – references are provided to Figures S5, S6 and tables S6, S7, which are not in the materials

Supplementary materials have been re-uploaded.

  1. Line 297 - you should select one section name

It has been corrected.

  1. Line 315 – a link is provided to Figure S7, which is not in the materials

Supplementary materials have been re-uploaded.

  1. Line 326-327 – a link to table S8 is provided, which is not in the materials

Supplementary materials have been re-uploaded.

  1. Unreadable captions in Figure 6b

We have improved it.

  1. Lines 351, 356, 362, 380-381 – references are given to Figures S9a-f, which are not in the materials

Supplementary materials have been re-uploaded.

  1. Line 416 – a link to table S13 is provided, there is no table itself

Supplementary materials have been re-uploaded.

  1. Line 430 – a link to table S14 is provided, there is no table itself

Supplementary materials have been re-uploaded.

  1. Line 529 - Can Fusariumand Penicillium be classified as bacteria? Is there a typo here?

Thank you for pointing out this error. Fusarium and Penicillium belonged to fungi. We have removed the word "bacteria".

Reviewer 2 Report

Comments and Suggestions for Authors

The writing has problems

1) far too many abbreviations 

2) too little justification    are any of the concentrations of OA used appropriate

3) unclear  most of the time what is being assayed     -  no supplementary material was available

4) you see changes but no conclusions made

changes may mean loss of some microbes  but the establishment of a well ajusted microbiome for the plant  

sticky notes illustrates the many places where clarity was needed for this reviewer

Comments on the Quality of English Language

in many places  i was unsure of what was being said  sorry    

Author Response

Thank you for the reviewer's suggestions. We have improved it.

1) far too many abbreviations.

Thank you for your suggestions. The full name of abbreviations has been added when all abbreviations first appeared in the manuscript.

2) too little justification are any of the concentrations of OA used appropriate.

Sorry about this problem. Due to a system error, the attached materials are not displayed. Additionally, Organic acid concentrations >250 mg/kg reduced the fresh weight of Sanqi (S1). Therefore, we determined that the continuous cropping obstacles of Sanqi occurred under three types of organic acids >250 mg/kg. Subsequently, we further analyzed the effects of organic acids >250 mg/kg on the relationship between soil conditions, microorganisms, and soil metabolites.

3) unclear most of the time what is being assayed. no supplementary material was available.

Sorry about this problem. Due to a system error, the attached materials are not displayed. We have re-uploaded the latest supplementary materials.

4) you see changes but no conclusions made changes may mean loss of some microbes but the establishment of a well ajusted microbiome for the plant.

Phthalate acid, palmitic acid, and mixed organic acid decreased the amount of beneficial (Badyrhizobium) and pathogenic bacteria (Xanthobacteriaceae, Ilyonectria) but increased the amount of other pathogenic (Fusarium) and harmful (Penicillium). The organic acid concentrations >250 mg/kg also decreased the complexity and stability of the bacterial and fungal network in the soil. These all indicate a decrease in soil nutrient content and a deterioration in plant growth.

Sticky notes illustrates the many places where clarity was needed for this reviewer.

1. Sanqi is used both as a name for plants and soil very confusing.

The name "Sanqi" refers to a plant name, and was widely used in other literature.

2. Why were these acids chosen? Whywould they be in these soils?

According to our previous study (Hei et al., 2023), phthalic and palmitic acids were mainly autotoxins and derived from the root exudates of Sanqi. We have added the sentence to explain the reason in line 17 of the manuscript.

3. What do these abbr mean?

We have added full names to all abbreviations when these first appeared in the manuscript. We have revised in line 20.

4. Where are these communities?phylosphere or where?

Thanks for your suggestion. We have added the words "in Sanqi soils" in line 23.

5. What organism?

We have modified it in line 25.

6. This is a general term you used very specific OA.

Thanks for your suggestion. We have revised it.

7. It still is not clear that these OA are native. Please document better. Why were these acids chosen.

We have improved it.

8. Really do you mean aromaticOA should see others citrate malate oxalate etc?

Organic acids (p-hydroxybenzoic, syringic, ferulic, p-coumaric, vanillic, and benzoic acid) have been identified in the prior studies, which is harmful to Sanqi growth and further result in Sanqi continuous cropping obstacles. Other organic acids remain unclear.

9. This justification should be in abstract.

We have added the reference to explain why the types of organic acids were selected in our study.

Hei, J.Y.; Wang, S.; He, X.H. Effects of exogenous organic acids on the growth, edaphic factors, soil extracellular enzymes, and microbiomes predict continuous cropping obstacles of Panax notoginseng from the forest understorey. Plant Soil. 2023, 1–18. https://doi.org/10.1007/s11104-023-06044-0.

10. Contents of what

We have revised it.

11. But those that remain are compatible.

We have improved it.

12. use this term throughout

We have modified it.

13. Do not understand that this must mean deterioration are these allopathic?

Yes, it is. This is an opinion based on the literature.

14. How were the concentrations chosen?

We have modified it in the line 103~108 of the introduction.

15. What is the growth matrix pH DOC etc?

It has been added in line 120 of the manuscript.

16. Need a control with water and ethanol is this your 0 value please confirm in text.

We have modified it in the line 124~127.

17. of what? how obtained?

We have modified it in line 173.

18. Was this the whole plant or what?

The whole plant of Sanqi. We have modified it.

19. What is CK?

CK is the control. The full name has already been used on lines 131 and 220 of the manuscript.

20. But look below muddled or was PA the only OA that caused an effect.

The two paregraphs showed the results of α- and β-diversity respectively.

21. was PA then the only active OA?

To express our meaning more accurately, we have changed "other treatments" to "phthalic acid".

22. what do colors mean cannot read x axis labels.

We have modified it.

23. of what?   

We have modified it.

24. I am lost with all of your abbreviations.

We have modified it.

25. no idea if this is a feasible concentration for the native soils.

It is feasible.

26. PA is associated with healthy plantscomment?

The palmitic acid rather than other organic acids treatment significantly alters the 117 DMs.

27. SEM? scanning electron microscopy?

It is the structural equation model (SEM) in line 415.

28. of what?

We have modified it.

29. not a bacillus

Fusarium and Penicillium belong to fungi. We have removed the word "bacteria".

Reviewer 3 Report

Comments and Suggestions for Authors

My opinion the manuscript requires additional professional evaluation by a microbiologist. In the review, I did not refer to microbial communities, but concentrate on the quality of the study. In general, there are so many methodological and editorial comments listed below:

Please explain in more detail term "continuous cropping obstacles" - lines 34-35

Line 105 Seeds or plants of Panax notoginseng obtained from forest?

It requires explanation of how plots of size (10 x 10 m) were designated when the width of the ridge at the bottom was from 120 to 150 cm.

Line 114 It was not specified at what development stage the Panax notoginseng transplant was at ridges planting.

Lines 120-121. No specify organic acid concentration.  The amount is given in mg per kg without indicating whether it is water or soil.

The concentration is given in mg per kg but the dose of the  in ml per plant (line 125).

In what growing stage of Panax notoginseng organic acids applied? (line 125)

What does the term Sanqi soil mean? (line 125)

Citation of source information regarding methods used in research should be in the references chapter and not in the methodology chapter. Too often it's used  (lines 154, 155, 176, 193, 195).

Lack of supplement file.

Figure 1. No explanation  of what the letters from "a" to "f" mean.

Scientific name names should be in Italics.

Badyrhizobium or Bradyrhizobium

Figure 6. No explanation  of what the letters from "a" to "c" mean. 

Author Response

Thank you for your suggestion, and we have revised it.

1. Please explain in more detail term "continuous cropping obstacles" - lines 34-35.

We have added it.

2. Line 105 Seeds or plants ofPanax notoginseng obtained from forest?

It has been corrected.

3. It requires explanation of how plots of size (10 x 10 m) were designated when the width of the ridge at the bottom was from 120 to 150 cm.

We have added the sentence "Each plot (10 m×10 m) included spaces of about 1m between the ridges." in line 142.

4. Line 114 It was not specified at what development stage the Panax notoginsengtransplant was at ridges planting.

"Sanqi roots during the dormancy period" have been added in lines 131-132.

5. Lines 120-121. No specify organic acid concentration. The amount is given in mg per kg without indicating whether it is water or soil.The concentration is given in mg per kg but the dose of the  in ml per plant (line 125).

It has been modified in line 140.

6. In what growing stage ofPanax notoginseng organic acids applied? (line 125).

We have added the growing stage in line 147.

7. What does the term Sanqi soil mean? (line 125)

Sorry, it is our negligence and should be the roots of Sanqi.

8. Citation of source information regarding methods used in research should be in the references chapter and not in the methodology chapter. Too often it's used  (lines 154, 155, 176, 193, 195).

We refer to other literature.

9. Lack of supplement file.

Sorry about this problem. Due to a system error, the attached materials are not displayed. We have resubmitted the supplementary materials.

10. Figure 1. No explanation of what the letters from "a" to "f" mean.

Thank you for the suggestion. We have improved the figure and the meaning of the letters.

11. Scientific name names should be in Italics. Badyrhizobium or Bradyrhizobium.

We have improved it.

12. Figure 6. No explanation of what the letters from "a" to "c" mean.

We have improved the figure and the meaning of the letters.

Round 2

Reviewer 2 Report

Comments and Suggestions for Authors

I am not sure whether the paper i reviewed (see attached) is your revision

some changes that you mention in your cover as being changed are not changed

the work continues to raise questions   in my mind  and so have made those comments

My main concern is that there is no clear discussion of  what the sampling replication was for the study    this should be clear in text   and in methods 

amount  eg 1 g soil etc is mentioned  only   THIS MUST BE RECTIFIED 

and if only single samples   WHAT IS THE VALIDITY OF ANY DATA set

my own experience with metabolites and microbial identity is the high  level of  variability   we started with about 180 metabolites but went down to only 30 or so that were consistent over   3 separate trials each with 3 replicates/sample   

But  the approach is good   

Comments on the Quality of English Language

many of these sticky notes are editorial   some sentences are not clear

use of caps  etc

Author Response

Dear reviewer:

We have tried our best to modify the article according to your suggestions. The specific location of your suggestions in the PDF version may have moved due to a problem with the PDF version. If any modifications have not been finished accurately, please help us to modify and improve; thank you very much.

In addition. All treated soil samples included 3 replicates, which we mentioned in the Materials method. At the same time, it can also be proved by uploading the sequencing results of the database. The sequence data have been deposited in the National Center for Biotechnology Information Sequence Read Archive under Accession no. PRJNA860751 (16S rRNA data) and no. PRJNA861133 (ITS data).

Reviewer 1

1.Why just promote pathogenic bacteria also fungi?

The manuscript's meaning is pathogenic microbes, including bacteria or fungi. We have modified it in the manuscript.

2.Difference in rhizosphere microbiomes?

Yes, the types of organic acid differ between conventional and organic management of Sanqi.

3.or growth environments

What we mainly emphasize is the different planting patterns.

4.soils growing poplar

We have modified it in line 77, as you suggested.

5.probably not the best word to start a paragraph   suggests should be in former paragraph

Thank you. We have merged this paragraph into the previous one.

6.as a mixture

We have modified it.

7.into soil?

We have modified it in line 118.

8.no cap O

We have modified it in line 96.

9.are pine needles major sources of these phenolics as they breakdown

Two organic acids (Phthalate and palmitic acid) are mainly derived from the root exudates of Sanqi.

10.rhizosphere of what the herb or pinus

We have modified it in line 177.

11.do you think your time period is sufficient to see  adaptability of one type of microbe to the acids?

Most exogenous organic acids can be absorbed or degraded in the soil in less than 15 days (Li et al., 2010). however, our experiment involved in 15days and 30 days.

Li, P.D., Wang, X.X., Li, Y.L., Wang, H.W., Liang, F.Y., Dai, C.C. The contents of phenolic acids in continuous cropping peanut and their allelopathy. Acta Ecol Sin. 2010, 30, 2128–2134.

12.really surprizing since the dose 250 seems to be effective in your previous discussions.

It is a error, which  have been modified in the manuscript.

13.again sorry.   rhizosphere soils for these assays?

Our experiments (microbes, soil metabolites) were all analyzed using the rhizosphere soil of Sanqi. For a more precise expression, we have modified it in line 320.

14.these are the formulae   not their analysis ie metabolites detected in analyses

Thank you. We have modified it.

15.present in the soils do not not have the pathways

We have modified it.

16.is had or has better   

It's had.

17.explain what you mean by plants? not clear from figure alone

It is the fresh weight of plants. We have modified Figure 7.

18.so is the major effect of the OA  to reduce pH?

Yes, it is. The pH value decreased at concentrations of <250 mg/kg.

19.not an easy sentence     not sure what you mean

We have modified it in line 431.

20.what do you mean by elements? do you mean essential metalsthe OAs you refer to are metal chelators

It refers to soil nutrients. We have modified it in line 433.

21.but this seems good  less denitrification?

We did not measure the denitrification rate.

22.or have major effects on the associations of the We have modified it, as you suggested.

23.i think its the other way round microbes are demineralizers   they cause dissolution of a range of soil minerals  and that shifts the soil physiochem

The interplay of plants, soil microbes, and soil physicochemical properties is complex. However, according to the results (SEM), the organic acids changed the edaphic factors, consequently affecting bacteria and fungi.

24.change pH and OA

According to the literature, many pine needles altered the soil’s nutrient content and increased microbial abundance.

25.these are not bacteria    you said you altered

We have modified it in line 487.

26.these  can have major effects on microbes

Yes, these can have significant effects on microbes. There are some literature has proved that saponins (ginsenosides Rg1, Rb1; notoginseng saponins R1; and ginsenosides Rh1) can inhibit root cell viability and seedling growth, leading to root rot (Gong et al., 2015; You et al., 2009).

Gong, J.T.; Cheng, X.Y.; Sun, M.; Wu, L.J.; Zhang, Z.L. Effects of three kinds of saponins on seed germination and seedling growth of Panax notoginseng. Acta Agric. Univ. Jiangxiensis. 2015, 37, 988–993. https://doi.org/10.13836/j.jjau.2015150.

You, P.J.; Zhang, Y.; Wang, W. Q.; Zhang, Z. L.; Yang, J. Z.; Yin, L. M. Allelopathic effects of continuous cropping soil of Panax Notoginseng on seed and seedling of some vegetables. Mod. Chin. Med. 2009, 5, 12–13. https://kns.cnki.net/kcms2/article/abstract?v=_NV5_o307ByDt7HJyk94Jy6RPQg0W-.

27.strange argument can you clarify i think you are trying to say Sanqi  releases OAs that are inhibitory to its own growth  as well as causing changes to the rhizosphere microbiome.

We have modified it, as you suggested.

28.caps again

We have modified it in the manuscript.

29.???  not in context

We have already mentioned it in line 362.

30.what plants Pinus or Sanqi

It is the fresh weight of Sanqi plants. We have already mentioned it in line 549.

31.how do you know they are pathogens   and pathogens to what a genus may have both  nonpathogenic and pathogenic species and the pathogenicity may change with respect to health of the plant if that is the hose  

Three organic acids increased the amount of other pathogenic (Fusarium) and harmful microbes (Penicillium). Some literature has proved that they cause plant wilt (Zhao et al., 2018) and generate toxic compounds (mycotoxins) (Perrone et al., 2017), respectively.

Zhao, Y.M.; Cheng, Y.X.; Ma, Y.N.; Chen, C.J.; Xu, F.R.; Dong, X. Role of phenolic acids from the rhizosphere soils of Panax notoginseng as a double-edge sword in the occurrence of root–rot disease. Molecules 2018, 23, 819. https://doi.org/10.3390/molecules23040819.

Perrone, G.; Susca, A. Penicillium species and their associated Mycotoxins. Methods Mol Biol. 2017, 1542, 107–119. https://doi.org/10.1007/978-1-4939-6707-05.